# Driving Factors Influencing the Decision to Purchase Plant-Based Beverages: A Sample from Türkiye

**DOI:** 10.3390/foods13111760

**Published:** 2024-06-04

**Authors:** Murat Baş, Meryem Kahriman, Gamze Ayakdas, Ladan Hajhamidiasl, Selen Koksal Koseoglu

**Affiliations:** 1Department of Nutrition and Dietetics, Faculty of Health Sciences, Acibadem Mehmet Ali Aydinlar University, Istanbul 34752, Türkiye; murat.bas@acibadem.edu.tr (M.B.); gamze.ayakdas@acibadem.edu.tr (G.A.); selen.koksal@acibadem.edu.tr (S.K.K.); 2Department of Nutrition and Dietetics, Institute of Health Sciences, Acibadem Mehmet Ali Aydinlar University, Istanbul 34752, Türkiye; ladan.hajhamidiasl@live.acibadem.edu.tr

**Keywords:** plant-based, beverage, actual buying behavior, attitude, environmental protection

## Abstract

In recent years, the trend toward plant-based beverages has continued to grow rapidly. This study aimed to assess the effects of sociodemographic characteristics and knowledge about plant-based beverages, subjective norms, perceived price, environmental protection, animal welfare, availability, and trust on attitudes and buying behavior toward these products. This study was conducted online using a two-part questionnaire prepared by considering the literature. This study included 935 participants, and our findings confirmed that the variable of environmental protection affects the attitude toward these products (β= 0.095; *p* = 0.007). Furthermore, gender, income level, lactose intolerance, and bloating due to cow’s or sheep’s milk influenced actual buying behavior (*p* < 0.05; *p* < 0.001). These findings indicate that people’s increased environmental protection awareness will positively influence attitudes towards plant-based beverages and that individuals who do not experience lactose intolerance and bloating due to cow’s or goat’s milk will have lower actual buying behavior. It was also determined that individuals with lower incomes bought more plant-based beverages. In conclusion, plant-based beverage marketers need to take into account individuals’ sociodemographic characteristics and environmental protection awareness when planning their marketing strategies.

## 1. Introduction

Plant-based beverages are defined as beverages created by breaking down plant material extracted in water and homogenizing this liquid, which resembles bovine milk in appearance [1,2]. These products are categorized into the following six different classes: cereal-based, legume-based, nut-based, oilseed-based, pseudo-cereal-based, and alternatives obtained from foods, including potatoes and moringa [3,4]. The products formed by this process are generally defined as drinks, beverages, or dairy alternatives instead of milk [5].

Over time, cow’s milk has been a significant part of human nutrition [6,7]. However, recently, with increasing urbanization and globalization, new and healthy food alternatives have been discussed to meet the changing needs and consumer demands. Thus, plant-based beverages have taken their place in the beverage industry [2]. The global dairy alternatives market is valued at USD 29.18 billion in 2023 and is expected to grow at a compound annual growth rate of 12.6% from 2023 to 2030 [8]. In Türkiye, 18% of consumers do not consume animal-based milk, whereas 45% consume plant-based beverages and foods [9]. In a study conducted in Türkiye with 796 female and 99 male participants, 36.4% of females and 72.7% of males did not consume plant-based dairy alternatives at all. Believing that plant-based dairy alternatives are beneficial to health increases the likelihood of consuming these products regularly [10]. Similarly, in the literature, health problems including lactose intolerance and cow’s milk allergy are effective in the trend toward plant-based dairy alternatives [11,12]. Along with health goals, the transition to a plant-based diet is also perceived as an important goal for the sustainability of the global food supply [13,14,15]. As animal products have higher water and ecological footprints, these products are one of the essential greenhouse gas producers [16]. Besides the sustainability impact, plant-based beverages are preferred as more affordable options in regions where mammalian milk is insufficient and expensive [12]. Conversely, the popularization of vegan nutrition as a healthier diet, increased awareness of ethics and rights, and especially the increase in animal rights advocates have supported the increase in the trend toward plant-based beverages [17].

Nutritional values, as well as taste, texture, or other delicate characteristics of plant-based beverages compared with animal milks, are essential to the trend of these products. Nutritional values can significantly vary depending on the raw material, production, added vitamins, and other ingredients such as flavoring or fats/oils [12]. Although these products are characterized by insufficient protein content and low mineral and vitamin bioavailability, the content of phenolic compounds, unsaturated fatty acids, antioxidant activity, and bioactive compounds such as phytosterols and isoflavones make them a good choice [18].

Determining the factors influencing the purchase of plant-based dairy alternatives is significant as these factors can direct the industry and market of these products. Therefore, this study aimed to examine the effects of sociodemographic factors, including age, gender, income level, marital status, and household size, on actual buying behavior for plant-based beverages. Moreover, this study aimed to evaluate the effects of knowledge of plant-based beverages, subjective norms, perceived price, environmental protection, animal welfare, availability, and trust on attitudes toward organic foods.

### 1.1. Literature Review and Hypotheses

Previous studies in the literature have presented data on the increasing demand for plant-based dairy products and the reasons for this demand [6,19,20,21,22]. The frequently reported reasons for this demand include sustainability [13,15], popularization of the vegan diet [17], increased ethics and rights awareness [17], inadequate availability of mammalian milk [12], and the health effects of plant-based beverages [11,12].

In this study, considering the motivational factors examined by Haas et al. [6] for plant-based beverage consumption and the variables identified by Gundala and Singh [23] and Singh and Verma [24] for organic food consumption, we determined the following hypotheses.

#### 1.1.1. Health Consciousness

Health consciousness refers to the tendency of individuals to take health-related actions and reflect these in their own behavior [25]. The fact that plant-based beverages have lower saturated fat content than mammalian milk and contain functionally active components supports them as a health option [2,26,27]. Additionally, these alternatives have become more essential owing to health concerns, including lactose intolerance and cow’s milk allergy [27]. In this regard, previous studies have reported that health consequences are effective in selecting these products. That is, individuals may be more likely to buy these products if they have a cow’s milk allergy or lactose intolerance, or if they think plant-based drinks are good for their health [10,18,28]. Therefore, we hypothesized the following:

**H1:** *Consumers’ health consciousness affects their attitudes toward buying plant-based beverages*.

#### 1.1.2. Knowledge

Consumers’ knowledge is a significant issue in explaining their buying behavior [29]. Knowledge is classified into the following two categories: objective and subjective knowledge [30]. Objective information refers to consumers’ accurate knowledge about a product, whereas subjective information refers to their perceptions of that product and its qualities. In other words, subjective knowledge is personal sensations about a product, and objective knowledge is the actual information about that product [31]. Consumers’ knowledge about a product can affect their purchasing attitudes [32]. A study conducted in the United Kingdom with 101 participants reported a lack of knowledge among consumers regarding plant-based beverages, with 42 male participants responding “No idea what nutrients in them are” [33]. Based on these findings, we hypothesized the following:

**H2:** *Consumers’ knowledge about plant-based beverages affects their attitude toward buying plant-based beverages*.

#### 1.1.3. Subjective Norm

Ajzen defined the concept of subjective norm as “perceived social pressure to perform or not to perform the behavior” [34]. In other words, this concept refers to the opinions of individuals who have an impact on an individual’s performance of a behavior and who have influence on his/her decision-making [35]. Previous studies have discussed the effects of subjective norms on consumers’ food and beverage choices [36,37]. A study examining the factors affecting organic food purchase reported that subjective norms affected the purchase intention through attitude formation [37]. However, other studies reported that subjective norms do not significantly affect the purchase intention of plant-based alternatives [38,39]. Considering these findings, we hypothesized the following:

**H3:** *Subjective norms affect consumers’ attitudes toward buying plant-based beverages*.

#### 1.1.4. Perceived Price

Price is an essential factor that can influence attitude and purchase intention. Plant-based beverages have generally higher prices than animal-based products [6]. Regarding this issue, Pritulska et al. [40], in their study examining consumer preferences for plant-based beverages with 436 participants, reported that the prices of these products were high and that sales would increase if the prices decreased. Accordingly, we hypothesized the following:

**H4:** *Perceived price affects consumers’ attitudes toward buying plant-based beverages*.

#### 1.1.5. Trust

Consumer trust refers to the confidence or belief that individuals have in products. The literature has proposed different classifications for consumer trust. This classification is based on different analytical levels (inter-personal or inter-organizational), the prevalence of single dimensions of trust across various levels (cognitive, emotional, or behavioral), and various levels of consistency [41]. Consumer trust can influence individuals’ buying behavior towards a product and is a prerequisite for market creation [42]. A previous study suggested that bias toward plant-based alternatives can affect the likelihood of purchasing these products [43]. In this direction, we hypothesized the following:

**H5:** *Trust affects consumers’ attitudes toward buying plant-based beverages*.

#### 1.1.6. Environmental Protection

Food production processes are one of the most significant factors leading to global environmental changes. Replacing animal-based foods with plant-based foods can minimize negative changes [14,44]. This situation may also affect consumers’ attitudes toward plant-based dairy products. Previous studies reported that consumers consider plant-based alternatives to be more sustainable than dairy products and may turn to plant-based alternatives owing to environmental concerns [15,33]. Considering the literature, we hypothesized the following:

**H6:** *Consumers’ thoughts about protecting the environment affect their attitudes toward buying plant-based beverages*.

#### 1.1.7. Animal Welfare

Animal welfare claims are becoming increasingly essential in milk and dairy product consumption [45]. Animal welfare-related certifications can be effective in reducing these concerns [46]. Consumers’ ability to obtain information about animal breeding conditions has a positive effect [46,47]. Regarding this issue, Boaitey and Minegishi [48] reported that participants who were concerned about animal welfare and the environment purchased plant-based beverages more frequently. Jiang et al. [49] suggested that animal welfare can be an essential external factor in consumers’ hedonic and emotional responses to milk. Accordingly, we proposed the following hypothesis:

**H7:** *Consumers’ thoughts about animal welfare affect their attitudes toward buying plant-based beverages*.

#### 1.1.8. Availability

Plant-based beverages are an increasing trend and can serve as an inexpensive alternative in regions with inadequate access to cow’s milk [2]. However, the purchasing decision is negatively affected by the insufficient availability of these products [50]. Therefore, we proposed the following hypothesis:

**H8:** *Availability affects consumers’ attitudes toward buying plant-based beverages*.

#### 1.1.9. Purchase Intention and Actual Buying Behavior

Bagozzi and Burnkrant defined purchase intention as “consumers’ subjective tendencies to purchase a product”. Consumer attitude is defined as the tendency to respond to a product’s cognitive, sensory, and behavioral stimuli [51]. Therefore, the food industry and market must consider the intent and attitude dimension for predicting the sales of products [52]. The literature has reported that several factors can affect purchase intention. These factors include individuals’ knowledge of the alternatives, their positive opinions, nutritional ingredients, health benefits, and sustainability [17,53,54,55]. Additionally, consumers’ intentions can predict their buying behavior and provide insight into actual buying behavior. However, purchase intention may not always turn into actual buying behavior [56]. Accordingly, studies on organic foods emphasized that consumers’ attitudes affect their purchase intentions and the effect of intention on actual buying behavior [23,24]. This is also imperative for plant-based beverages. Therefore, we hypothesized the following:

**H9:** *Consumers’ attitudes toward plant-based beverages affect their purchase intentions*.

**H10:** *Consumers’ purchase intentions regarding plant-based beverages affect their actual buying behavior*.

**H11:** *Consumers’ attitudes affect actual buying behavior through the mediator effect of purchase intention*.

#### 1.1.10. Sociodemographic Factors

In addition to consumers’ attitudes, sociodemographic factors are also significant indicators affecting buying behavior. One factor that drives plant-based beverage consumption is age. In their study examining consumers’ motivations for buying dairy products and plant-based beverages, Boaitey and Minegishi reported that the younger generation is more likely to consume plant-based beverages [48]. Conversely, Palacios et al. stated that age did not have a significant effect on the general liking for soy beverages [57]. Besides age, studies in the literature highlighted that consumers of plant-based alternatives are generally females [58,59]. These findings highlighted the significance of gender. Considering the high prices of plant-based beverages, consumers’ income levels have also become a significant factor affecting their buying behavior. Consistent with this, Slade reported that individuals who buy plant-based dairy products generally have a high income [60]. Moreover, the education level of consumers has been focused on in the literature, and increasing the education level can positively affect consumers’ tendencies toward these alternatives [60,61].

Health concerns of consumers may also be effective in turning to plant-based beverages. Particular emphasis is placed on the impact of individuals’ diagnoses, including lactose intolerance and cow’s milk allergy, on this trend [2,3]. Furthermore, plant-based alternatives can be preferred to avoid hypercholesterolemia as they have lower saturated fat and cholesterol contents [28].

A study with 896 participants conducted to evaluate the factors affecting the purchase of plant-based beverages in Türkiye stated that sociodemographic factors including age, monthly income, and education level affected the purchasing decision. However, consumers’ preferences for plant-based beverages were not affected by the presence of lactose intolerance [10]. Accordingly, we hypothesized the following:

**H12a:** *There is a significant difference in the actual buying behavior of consumers for plant-based beverages according to the gender variable*.

**H12b:** *There is a significant difference in the actual buying behavior of consumers for plant-based beverages according to the age variable*.

**H12c:** *There is a significant difference in the actual buying behavior of consumers for plant-based beverages according to the education variable*.

**H12d:** *There is a significant difference in the actual buying behavior of consumers for plant-based beverages according to the occupation variable*.

**H12e:** *There is a significant difference in the actual buying behavior of consumers for plant-based beverages according to the marital status variable*.

**H12f:** *There is a significant difference in the actual buying behavior of consumers for plant-based beverages according to the income variable*.

**H12g:** *There is a significant difference in the actual buying behavior of consumers for plant-based beverages according to the presence of the lactose intolerance variable*.

**H12h:** *There is a significant difference in consumers’ actual buying behavior of plant-based beverages depending on whether they experience bloating when drinking cow’s or sheep’s milk*.

**H12i:** *There is a significant difference in consumers’ actual buying behavior of plant-based beverages depending on whether they have been diagnosed with a chronic disease in the last year*.

**H12j:** *There is a significant difference in the actual buying behavior of consumers for plant-based beverages according to the household size variable*.

## 2. Materials and Methods

### 2.1. Data Collection Procedure

This study, which was conducted between April 2023 and March 2024, was conducted online with a questionnaire prepared with Google Forms. The questionnaire was shared with the participants through social media platforms. Participants aged 18 years or older and who volunteered to participate were included in this study. Those without Turkish reading and comprehension skills were not included in the study. The number of participants to be included in this study was calculated as 170, with a = 0.05 margin of error and 95% test power using the G*Power program (version number 3.1), considering the studies in the literature [6,19,20]. This study was approved by the Acibadem Mehmet Ali Aydinlar University Medical Research Ethics Committee (ethical approval number: ATADEK 2024-3/124). Moreover, all participants provided informed consent, and the processes were conducted in accordance with the Declaration of Helsinki. Ultimately, 935 participants were reached.

A two-part questionnaire was used to test the hypotheses (Appendix A). While preparing the survey, studies in the literature were reviewed [6,19,20]. The first part included questions examining the sociodemographic characteristics and general nutritional habits of the participants. Additionally, in this section, the participants were asked about their plant-based dairy product consumption. The second part comprised three different sections. All items in this part were scored on a 5-point Likert-type scale ranging from “strongly disagree” to “strongly agree”. The first section asked about factors affecting consumers’ plant-based beverage consumption. The second section asked about consumers’ preferences for plant-based dairy products. The final section questioned about factors including brand, taste, and price that may affect consumers’ decision to purchase plant-based dairy products.

### 2.2. Statistical Analysis

Descriptive statistics for categorical variables were presented as frequencies and percentages. The Shapiro–Wilk test was used to examine the conformity of numerical variables to the normal distribution. Descriptive statistics of numerical variables were expressed as means ± standard deviations (X ± SD) for normally distributed data, and median (min–max) values for non-normally distributed data.

Convergent validity was considered for the construct validity of the questionnaire. Convergent validity means that the items related to the variables express that they are related to each other and to the factor they form. For convergent validity, all composite reliability (CR) values of the scale are expected to be greater than the average variance extracted (AVE) values, and the AVE value is expected to be >0.5. Additionally, the standardized factor loadings of the items must be above 0.5, and the CR value must be higher than 0.7 [62]. To determine the reliability level of the questionnaires, the Cronbach’s alpha coefficient was calculated.

For discriminant validity, the square of the maximum shared variance (MSV), the average of the square of the shared variance (ASV), and the AVE values were evaluated. The MSV value is the square of the highest variance that a factor shares with any of the other factors. The ASV value is obtained by dividing the sum of the squares of the variance shared by a factor with other factors by the number of shared variances. To mention discriminant validity, the conditions of MSV < AVE, ASV < MSV, and the square root of AVE must be greater than the correlation between factors [63].

Mediation analysis is a statistical approach used to test whether there is a role for another variable or variables when the relationship between two variables is known. Structural equation modeling was used for the mediation analysis in this study. Structural equation modeling analyses were performed using the R Project v3.6.1 software [64].

To examine relationships between the scales for normally distributed data, the Spearman correlation coefficient was used. Multiple regression analysis, which is based on the mathematical equivalence of the effect between two or more dependent variables, was used to test the relationship between variables. In all calculations and interpretations, the statistical significance level was considered “*p* < 0.05, *p* < 0.01, and *p* < 0.001”, and the hypotheses were established bidirectionally. All statistical analyses were performed using the Statistical Package for the Social Sciences v27 [65] and R Project v3.6.1 [64] package programs.

## 3. Results and Discussion

Descriptive statistics of the demographic, health, and plant-based beverage consumption findings of the participants are presented in Table 1. A total of 935 participants were included, of whom 69.9% (n = 654) and 30.1% (n = 281) were females and males, respectively. The mean age of the participants was 33.84 ± 11.82 years, 57.8% had a bachelor’s degree, 32.8% were students, 62.1% were single, 25% had an income level of TRY 20,001–30,000, and the mean number of individuals living in the household was 3.06 ± 1.28.

Of the participants, 56.7% did not have lactose intolerance, 46.3% did not experience bloating when drinking cow’s or sheep’s milk, and 18.7% had a new chronic disease diagnosed by a doctor in the last year. Furthermore, 97.1% of the participants defined their dietary pattern as omnivore.

Of the participants, 60.9% purchased plant-based beverages. The percentages of female and male participants purchasing plant-based beverages were 48.8% and 92.4%, respectively. Among the participants who purchased plant-based alternatives, 92.8%, 41.1%, 39.2%, 38.1%, 16.2%, 3.7%, and 2.6% preferred almond-, coconut-, oat-, soy-, hazelnut-, walnut-, and rice-based beverages, respectively (Table 1).

Analysis of the study consumers’ purchasing preferences revealed that 84.6%, 73.4%, and 67.8% of the participants preferred to purchase previously tasted plant-based beverages, plant-based beverages with promotions, and plant-based beverages without added sugar, respectively. Additionally, pasteurized, high-protein, low-fat, low-priced, low-calorie, calcium-enriched, and flavored products were preferred at lower rates (Table 2).

When the factors influencing the participants’ decision to purchase plant-based beverages were investigated, taste showed an effect at a rate of 100%, followed by health benefits (96.1%), from which plant it was made (95.3%), nutritional value (94.7%), and freshness (94.4%). Furthermore, price, brand, and whether there was a promotion or not affected the decision to purchase plant-based beverages (Table 2). Studies in the literature have also shown that several factors including nutrient content, diet, socioeconomic status, health effects, environmental protection, and sustainability influenced the preference and purchasing behavior for plant-based beverages [10,17,57,58,66].

Plant-based beverages have gained a lot of attention in recent years due to their tendency to be natural and environmentally friendly, with a similar taste to animal-based foods. The majority of modern consumers prefer to consume highly sustainable and environmentally friendly products. For instance, one study concluded that consumers of plant-based beverages are driven by a desire to reduce their intake of animal products, concerns about animal welfare, and a perception of a lower environmental impact compared to dairy milk [17]. At the same time, growing health consciousness and potential health benefits play an important role in shaping consumer attitudes towards plant-based beverages. One study reported that vegetarian, health-conscious consumers and consumers with high nutritional knowledge tend to purchase plant-based alternatives [39]. In a survey study on consumer acceptability of plant-based beverages and dairy products, many consumers’ reasons for choosing plant-based beverages were based on their intention to minimize their carbon footprint/GHG emissions, indicating sustainability [67]. Plant-based beverages have higher production costs and therefore higher prices than cow’s milk. As a result, plant-based beverages have a significant negative impact on consumer purchasing behavior due to their high prices [40].

Examining the analysis results (Table 3) showed that the standardized factor loadings of the items ranged from 0.567 to 0.967, CR and Cronbach’s alpha values were higher than 0.7, AVE values were >0.5, and all 11 constructs had convergent validity. Therefore, these results show that the scale used in this study has convergent validity and its construct validity is confirmed.

Investigating the discriminant validity of the study variables showed that the AVE values were lower than the corresponding CR values, and all ASV values were lower than the MSV values. Based on the findings, it was determined that discriminant validity was achieved (Appendix A).

As the scores of animal welfare and environmental protection among the study factors increased, attitude scores increased by 6.7% and 8.6%, respectively. A significant positive moderate correlation (*p* < 0.001) was noted between attitude scores and actual buying behavior scores. Examining the results revealed that as the attitude scores increased, a 53.7% increase in actual buying behavior scores was noted (Appendix A).

Analyzing the effects of the study variables on the attitude variable showed that the environmental protection variable had a significant direct effect on the attitude variable (*p* < 0.01). Analysis of the effect results showed that environmental protection scores had an effect of 9.5% on attitude scores (Table 4; Figure 1). Our results are similar to those of previous studies showing that consumers’ environmental protection awareness and sustainable food system motives influence their attitude toward purchasing plant-based beverages [15,33]. Thus, the literature and our results confirm H6.

A review of the literature shows that animal welfare is one of the main reasons why consumers avoid dairy and switch to plant-based beverages. The identified effects of subjective norms also suggest that negative animal welfare events can have both direct and indirect effects on dairy consumption [46]. At the same time, price perception and availability of plant-based beverages also influence buying behavior [50,66].

In a study, health awareness positively affects perceived knowledge and attitude. Moreover, perceived knowledge positively influences consumers’ attitude towards foods, which means that consumers’ health awareness and attitude towards foods are developed based on their perceived knowledge [31]. Previous studies have shown that consumers are reluctant to accept foods when they do not have enough information about nutrition information and the health benefits of foods, or when they find it difficult to understand [68]. In this context, some studies show that increasing consumers’ nutrition knowledge can change their dietary behavior. Thus, both health effects can be benefited from and a wider acceptance of foods can be ensured [69].

Compared to the literature, our results showed that the above-mentioned factors of health consciousness, knowledge, subjective norm, perceived price, trust, animal welfare, and availability did not have a significant effect on the attitude to purchase plant-based beverages in the sample in our study.

Our results showed that the attitude variable, which was mediated by the purchase intention variable, had no statistically significant effect (*p* > 0.05) on the actual buying behavior variable (Table 5; Figure 1). A study directly examining the relationship between plant-based beverages and attitude, purchase intention, and actual buying behavior was not noted in the literature. However, we examined results from similar studies on plant-based foods. A previous study concluded that consumers who have a positive attitude toward plant-based meat alternatives are more willing to purchase them, which positively affects their buying behavior [70]. Other studies on plant-based foods in the literature supported that a significant relationship exists between attitude and purchase intention [39,71]. Price and accessibility also influence consumers’ attitudes and buying behavior. Consumers generally prefer to buy affordable and accessible foods [72]. Since the production cost of plant-based beverages is higher, the price of plant-based milk substitutes is higher than cow’s milk. This may have a negative impact on buying behavior [57]. In our study, it is seen that a high price does not affect the buying behavior of consumers. In this case, although the literature confirms H9, H10, and H11, our results are inconsistent with the literature and these hypotheses.

Examining the effects of demographic variables of consumers on actual buying behavior scores revealed that age, education level, occupation, marital status, chronic disease status, and household size variables did not have a significant effect on actual buying behavior scores (*p* > 0.05).

There are contradictory results in the literature regarding the effect of the age factor on actual buying behavior. Although some studies have reported that age is effective and that the younger population prefers plant-based beverages more, some studies have observed that it has no significant effect on actual buying behavior [73], which is consistent with our findings.

A previous study concluded that individuals working in animal farms are more likely to consume dairy products, whereas individuals with university education are more likely to consume plant-based beverages. From this study, it can be concluded that education level and occupation have an indirect effect on the intention to purchase plant-based beverages [74]. However, our present study showed that education level and occupation did not have a significant effect on the actual buying behavior of plant-based beverages.

In the literature, a direct study on the effect of marital status, the presence of chronic disease in the last year, and household size on the buying behavior of plant-based beverages was not observed. However, studies showing that household size [48,75] and the presence of chronic disease [28] affect the food choice and consumption of plant-based beverages were noted. Our results suggest that these variables had no effect.

Conversely, our results showed that gender, monthly income level, lactose intolerance status, and bloating when drinking cow’s or sheep’s milk had a statistically significant effect on actual buying behavior scores (*p* < 0.05; *p* < 0.001). The actual buying behavior of males was 1.592-fold higher than that of females. Our results confirm the conclusion that gender affects the actual buying behavior, as in the results of the studies in the literature. However, in the literature, the actual buying behavior of females is higher, which is different from our results [58,59]. When it is assumed that all these sociodemographic factors that increase actual buying behavior are collected in the same individual, it is thought that buying behavior may increase with multiple effects.

Our results showed that the actual buying behavior of individuals with a monthly income level of TRY 20,001–40,000 was 0.754-fold higher than that of individuals with a monthly income level of TRY 40,001 and above. Thus, according to our results, individuals with a lower economic income show higher plant-based beverage buying behaviors, contradicting studies in the literature that conclude that individuals with a higher income buy more plant-based beverages [60].

Individuals who were not lactose intolerant had a 0.967-fold lower actual buying behavior than those with lactose intolerance. Similarly, individuals who did not experience bloating after drinking cow’s or sheep’s milk had a 2.424-fold lower actual buying behavior than those who experienced bloating. Investigating the literature revealed that plant-based beverages will be an alternative in nutrition in health conditions, such as lactose intolerance and cow’s milk protein allergy [76,77]. Conversely, studies showing that lactose intolerance is ineffective in the preference for plant-based beverages are noted [10]. Our results are also consistent with the literature. Our results showed that some, but not all, demographic factors influence the buying behavior for plant-based alternatives, confirming H12a, H12f, H12g, and H12h (Table 6; Figure 2).

Individuals’ purchase intentions and actual buying behavior for changing attitudes are depicted in Figure 3. Individuals with high purchase intentions had high actual buying behavior for their changing attitudes, whereas individuals with low purchase intentions had lower actual buying behavior for their changing attitudes. Individuals’ attitudes toward a product can be a significant determinant of their purchase intentions [51], and purchase intentions can result in actual buying behavior [78]. Related to this issue, a study conducted with 770 consumers in the United States confirmed that attitude positively impacts consumers’ purchase intention and actual buying behavior. Moreover, attitude and purchase intention mediate the relationship between influencing factors and actual buying behavior [23]. A previous study on organic food consumption in Türkiye emphasized that participants with high purchase intentions had higher actual buying behavior for changing attitudes, whereas those with low purchase intentions had lower actual buying behavior [78]. The literature and our findings suggest that the effect of purchase intention on actual buying behavior can vary for changing attitudes.

## 4. Conclusions

Our findings showed that gender, monthly income level, lactose intolerance status, and bloating experienced when drinking cow’s or sheep’s milk influenced actual buying behavior. However, this is not consistent with the literature, where lower-income individuals had higher buying behavior towards plant-based beverages. Considering that individuals generally prefer more accessible and affordable products, this finding is new and different from the literature. In addition, the environmental protection variable, one of the study variables, directly affected attitude. The findings of this study provide evidence to marketers that individuals’ attitudes towards these products and their actual buying behavior may be influenced by sociodemographic factors, health problems, and environmental concerns associated with dairy consumption. However, it is believed that by integrating environmental protection awareness, marketers of these products can present to the public that plant-based beverages are not a substitute for dairy products, but may be an alternative for some individuals with health problems or low income.

### Limitations

This study had some limitations. First, as the study was conducted online, all results were based on participants’ statements. Second, most of the study participants had a bachelor’s degree. Of note, different results may be obtained for participants with lower education levels. Lastly, the effects of only some variables were examined. There is, therefore, a need for larger-scale studies with more diverse samples and a wider range of variables to provide a comprehensive understanding of consumer behavior towards plant-based beverages.

## Figures and Tables

**Figure 1 foods-13-01760-f001:**
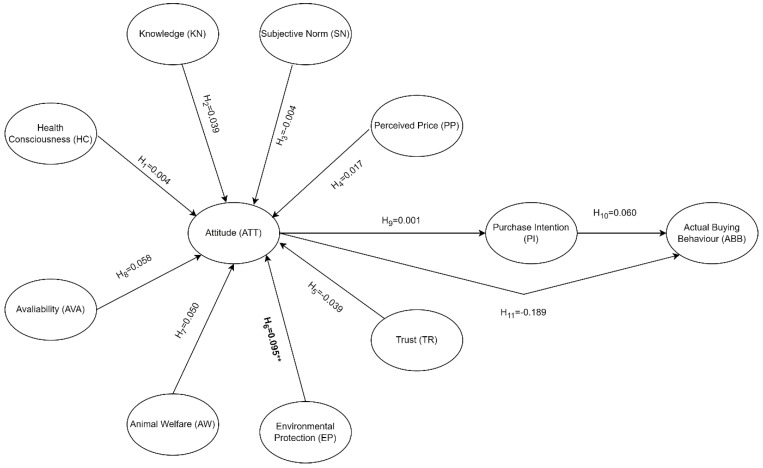
Model testing the effects of study variables on attitude, purchase intention, and actual buying behavior. ** indicates statistically significant *p*-value.

**Figure 2 foods-13-01760-f002:**
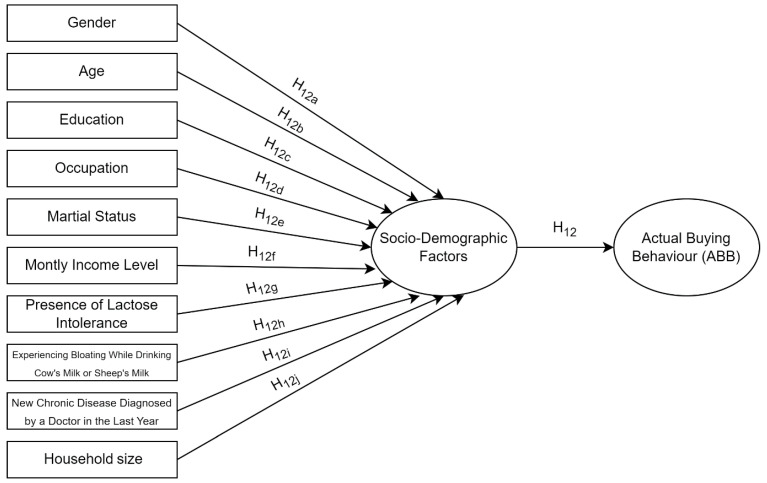
Multiple regression model evaluating the effects of sociodemographic factors on actual buying behavior.

**Figure 3 foods-13-01760-f003:**
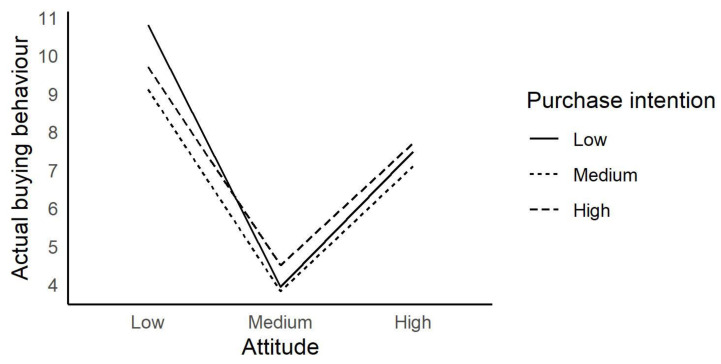
Regulatory effect of purchase intention on attitude and actual buying behavior.

**Table 1 foods-13-01760-t001:** Demographic, health, and plant-based beverage consumption findings of the consumers.

	Males (n = 281)		Females (n = 654)		Total (n = 935)	
	n	%	n	%	n	%
Age (year) ($\bar{X}\pm SS$)	34.57 ± 12.58		33.52 ± 11.48		33.84 ± 11.82	
Education level						
High school and lower education	78	27.8	146	22.3	224	24.0
Bachelor’s degree	157	55.9	383	58.6	540	57.8
Master’s and Doctorate	46	16.4	125	19.1	171	18.3
Occupation						
Civil servant	28	10.0	67	10.2	95	10.2
Private sector	88	31.3	180	27.5	268	28.7
Self-employment	36	12.8	59	9.0	95	10.2
Employer	29	10.3	16	2.4	45	4.8
Retired	11	3.9	23	3.5	34	3.6
Housewife	3	1.1	38	5.8	41	4.4
Student	79	28.1	228	34.9	307	32.8
Unemployed	7	2.5	43	6.6	50	5.3
Marital status						
Married	116	41.3	238	36.4	354	37.9
Single	165	58.7	416	63.6	581	62.1
Purchasing plant-based alternatives within the last month						
Yes, I regularly purchase plant-based alternatives.	125	44.5	181	27.7	306	32.7
Yes, I bought it once last month.	67	23.8	74	11.3	141	15.1
Yes, I bought it twice last month.	37	13.2	33	5.0	70	7.5
Yes, I bought it three times last month.	21	7.5	31	4.7	52	5.6
No, I did not purchase it.	31	11.0	335	51.2	366	39.1
Type of plant-based milk alternatives purchased *						
Almond-based beverage	231	92.4	297	93.1	528	92.8
Oat-based beverage	101	40.4	122	38.2	223	39.2
Soy-based beverage	95	38.0	122	38.2	217	38.1
Coconut-based beverage	100	40.0	134	42.0	234	41.1
Hazelnut-based beverage	35	14.0	57	17.9	92	16.2
Rice-based beverage	8	3.2	7	2.2	15	2.6
Walnut-based beverage	8	3.2	13	4.1	21	3.7
Monthly income level						
TRY 10,000 and below	40	14.2	84	12.8	124	13.3
TRY 10,001–20,000	29	10.3	147	22.5	176	18.8
TRY 20,001–30,000	38	13.5	196	30.0	234	25.0
TRY 30,001–40,000	63	22.4	152	23.2	215	23.0
TRY 40,001 and above	111	39.5	75	11.5	186	19.9
Lactose intolerance						
Yes	65	23.1	98	15.0	163	17.4
No	105	37.4	425	65.0	530	56.7
Not sure	111	39.5	131	20.0	242	25.9
Bloating after drinking cow’s or sheep’s milk						
Yes	141	50.2	187	28.6	328	35.1
No	72	25.6	361	55.2	433	46.3
Sometimes	68	24.2	106	16.2	174	18.6
Nutrition model						
Omnivore	273	97.2	635	97.1	908	97.1
Vegetarian	6	2.1	14	2.1	20	2.1
Flexitarian	2	0.7	5	0.8	7	0.7
New chronic disease diagnosed by a doctor in the last year						
Yes	49	17.4	126	19.3	175	18.7
No	232	82.6	528	80.7	760	81.3
Number of individuals living in the household ($\bar{X}\pm SS$)	3.06 ± 1.27		3.05 ± 1.29		3.06 ± 1.28	

*: More than one answer is provided.

**Table 2 foods-13-01760-t002:** Consumers’ plant-based alternative purchase preferences and factors influencing plant-based beverage purchasing decision.

	Strongly Disagree		Disagree		Undecided		Agree		Strongly Agree	
Purchase Preferences	n	%	n	%	n	%	n	%	n	%
I buy a plant-based alternative with no added sugar.	25	4.4	73	12.9	85	15.0	190	33.5	195	34.3
I buy a flavored plant-based alternative.	190	33.5	187	32.9	98	17.3	70	12.3	23	4.0
I buy a plant-based alternative enriched with calcium.	34	6.0	92	16.2	237	41.7	150	26.4	55	9.7
I buy a low-calorie plant-based alternative.	35	6.2	135	23.8	192	33.8	170	29.9	36	6.3
I buy a high-protein plant-based alternative.	17	3.0	82	14.4	163	28.7	200	35.2	106	18.7
I buy a low-fat plant-based alternative.	34	6.0	107	18.8	184	32.4	191	33.6	52	9.2
I buy pasteurized plant-based alternative.	12	2.1	58	10.2	171	30.1	254	44.7	73	12.9
I buy a low-priced plant-based alternative.	33	5.8	137	24.1	165	29.0	167	29.4	66	11.6
I buy a plant-based alternative that I have tasted before.	0	0.0	28	4.9	59	10.4	274	48.2	207	36.4
I buy a plant-based alternative with a promotion.	14	2.5	60	10.6	77	13.6	234	41.2	183	32.2
**Factors influencing purchasing decisions**										
Brand	0	0.0	8	1.4	55	9.7	187	32.9	318	56.0
Taste	0	0.0	0	0.0	0	0.0	156	27.5	412	72.5
Freshness	0	0.0	9	1.6	23	4.0	144	25.4	392	69.0
Nutritional value	0	0.0	0	0.0	30	5.3	182	32.0	356	62.7
What plant is it made from?	0	0.0	0	0.0	27	4.8	198	34.9	343	60.4
Price	15	2.6	17	3.0	66	11.6	167	29.4	303	53.3
Health benefits	0	0.0	0	0.0	22	3.9	158	27.8	388	68.3
Whether there was a promotion or not	33	5.8	68	12.0	105	18.5	166	29.2	196	34.5

**Table 3 foods-13-01760-t003:** Reliability and convergent validity.

Constructs	Items	Standardized Factor Loadings	CR	AVE	Cronbach’s Alpha
Health consciousness (HC)	HC1: I am aware of my health	0.885	0.931	0.730	0.907
	HC2: I am concerned about the type and amount of nutrition in the food that I consume daily	0.885			
	HC3: I think about my health	0.874			
	HC4: Plant-based beverages are healthier than animal milk	0.874			
	HC5: Plant foods are healthier than animal foods	0.746			
Knowledge (KN)	KN1: Plant-based beverages contain high amounts of protein	0.881	0.872	0.696	0.776
	KN2: Plant-based beverages contain low amounts of fat	0.857			
	KN3: Plant-based beverages contain high amounts of calcium	0.759			
Subjective norm (SN)	SN1: Many people who are important to me consider plant-based beverages to be healthy	0.914	0.903	0.757	0.839
	SN2: Many people who are important to me consider plant-based beverages to be environmentally friendly	0.901			
	SN3: Many people who are important to me pay attention to environmentally friendly nutrition	0.789			
Perceived price (PP)	PP1: It is too expensive for me to buy plant-based beverages	0.934	0.880	0.715	0.785
	PP2: I need to have more money to buy more healthy and environmentally friendly food	0.922			
	PP3: I prefer to buy a plant-based beverage, even if it is expensive	0.651			
Purchase intention (PI)	PI1: If I want to buy a plant-based beverage, I can buy it	0.956	0.938	0.793	0.910
	PI2: I intend to consume plant-based beverage alternatives in the future	0.956			
	PI3: I am always interested in buying more plant-based beverages for the individual or family’s needs	0.870			
	PI4: I’m thinking of purchasing a plant-based beverage next month	0.765			
Actual buying behavior (ABB)	ABB1: I have been a regular buyer of plant-based beverages	0.951	0.949	0.862	0.918
	ABB2: I still buy plant-based beverages even though conventional alternatives are on sale	0.918			
	ABB3: I never mind paying premium price for plant-based beverages	0.915			
Attitude (ATT)	ATT1: I believe a plant-based beverage is very useful to meet the nutritional needs	0.967	0.931	0.821	0.884
	ATT2: Plant-based milk beverages have higher quality than conventional ones	0.967			
	ATT3: I am convinced the consumption of plant-based beverages is a reasonable action	0.769			
Trust (TR)	TR1: I trust the label information of plant-based beverages	0.918	0.915	0.636	0.812
	TR2: I trust plant-based milk producers	0.918			
Environmental protection (EP)	EP1: Plant-based beverages are environmentally friendly	0.946	0.904	0.761	0.841
	EP2: Farm animals harm the environment	0.944			
	EP3: The environment is heavily exploited by humans	0.705			
Animal welfare (AW)	AW1: Animals must be kept in their natural habitat	0.923	0.887	0.669	0.828
	AW2: Companies must think about their profits, but also about animals	0.920			
	AW3: I care about the welfare of animals	0.811			
	AW4: I am worried about the animals on the farms for milk production	0.567			
Availability (AVA)	AVA1: I would like to find plant-based beverages in every grocery store and other shopping stores	0.911	0.912	0.775	0.851
	AVA2: I would like to find plant-based beverages in restaurants and cafes	0.897			
	AVA3: I would like to have plant-based beverages in online sales areas	0.831			

CR: composite reliability; AVE: average variance extracted.

**Table 4 foods-13-01760-t004:** Effects of study variables on attitude variable.

	β	t-Value	*p*-Value
HC→ATT	0.004	0.167	0.867
KN→ATT	0.039	1.058	0.290
SN→ATT	−0.004	−0.109	0.913
PP→ATT	0.017	0.489	0.625
TR→ATT	−0.039	−0.780	0.436
EP→ATT	0.095	2.709	0.007 **
AW→ATT	0.050	1.730	0.084
AVA→ATT	0.058	1.701	0.089

HC: health consciousness; KN: knowledge; SN: subjective norm; PP: perceived price; ATT: attitude; TR: trust; EP: environmental protection; AW: animal welfare; AVA: availability; β, beta coefficient; and ** *p* < 0.01.

**Table 5 foods-13-01760-t005:** The influence of attitude on purchase intention, purchase intention on actual buying behavior, and attitude as a mediator of purchase intention on actual buying behavior.

	β	t-Value	*p*-Value
ATT→PI	0.001	0.012	0.990
PI→ABB	0.060	0.532	0.595
ATT→PI→ABB	−0.189	−1.597	0.111

ATT: attitude; PI: purchase intention; ABB: actual buying Behavior; and β: beta coefficient.

**Table 6 foods-13-01760-t006:** Influence of consumers’ demographic variables on the actual buying behavior variable.

	Unstandardized Coefficients				95.0% Confidence Interval for β	
	β	SE	t	*p*	Lower Bound	Upper Bound
(Constant)	8.025	0.786	10.207	<0.001 ***	6.482	9.569
Gender (Ref: Women)						
Man	1.592	0.289	5.504	<0.001 ***	1.025	2.160
Age (year)	0.007	0.011	0.590	0.555	−0.016	0.029
Education (Ref: Bachelor and above)						
High school and below	0.159	0.382	0.416	0.678	−0.591	0.909
Associate degree	−0.269	0.320	−0.839	0.402	−0.897	0.360
Occupation (Ref: Worker)						
Nonworker	−0.232	0.268	−0.865	0.387	−0.757	0.294
Marital status (Ref: Married)						
Single	0.317	0.286	1.108	0.268	−0.244	0.877
Monthly income level (TRY 40,001 and above)						
TRY 20,000 and below	−0.254	0.356	−0.714	0.475	−0.952	0.444
TRY 20,001–40,000	0.754	0.336	2.247	0.025 *	0.096	1.413
Presence of lactose intolerance (Ref: Yes)						
No	−0.967	0.457	−2.116	0.035 *	−1.864	−0.070
Not sure	0.615	0.414	1.484	0.138	−0.198	1.427
Experiencing bloating while drinking cow’s or sheep’s milk (Ref: Yes)						
No	−2.424	0.391	−6.196	<0.001 ***	−3.192	−1.656
Sometimes	−0.124	0.390	−0.318	0.751	−0.889	0.641
New chronic disease diagnosed by a doctor in the last year (Ref: Yes)						
No	0.044	0.304	0.144	0.885	−0.553	0.641
Household size	−0.035	0.093	−0.381	0.703	−0.217	0.146

β: beta coefficient; SE: standard error; t: independent sample *t*-test; * *p* < 0.05; *** *p* < 0.001.

## Data Availability

The original contributions presented in the study are included in the article/Appendix A, further inquiries can be directed to the corresponding author.

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
