# Peer review of "Driving Factors Influencing the Decision to Purchase Plant-Based Beverages: A Sample from Türkiye"

_foods, 2024, doi:10.3390/foods13111760_

Round 1
Reviewer 1 Report
Comments and Suggestions for Authors
Adjusting the alignment of subtitles will ensure consistency and readability throughout the manuscript.
Please adjust the tables to have equal spacing on both the left and right sides for better alignment.
Please provide additional information in the title of Figure 1 and Figure 2.
Please ensure that the beta coefficients in the tables are aligned consistently with the numbers for improved readability and presentation.
Please remove any unnecessary spaces in the manuscript to improve readability and formatting consistency.
There's no need for excessive space before the references section.
There's no need for a title within the conclusion section.
References listed from 462 to 467 need to be made consistent with the formatting, spacing, and alignment of other references for uniformity and clarity.
The reference on line 581 needs to be aligned properly for consistency with the rest of the references section.
Please ensure consistency in the alignment of text within the references section.
Authors should consider adding space or lines to Table 2 to improve readability, particularly where numbers are closely positioned, making it difficult to differentiate between them. This adjustment will enhance clarity and make it easier for readers to follow the data presentation.
Abstract
Ensure that the abstract is clear, concise, and effectively communicates the main findings of the study.
Highlight the most important findings of the study, such as the significant factors influencing attitudes and buying behavior towards plant-based beverages.
Discuss the practical implications of the findings for marketers and other stakeholders in the plant-based beverage industry. This could include recommendations for marketing strategies or product development based on the identified factors influencing consumer behavior.
Introduction
Provide a concise definition of plant-based beverages at the beginning of the introduction to ensure clarity for readers who may not be familiar with the term.
Explicitly state the objectives of the study to guide readers on what to expect from the research. This could involve summarizing the aim of examining the effects of sociodemographic factors and other variables on attitudes and buying behavior towards plant-based beverages.
While some relevant literature is briefly mentioned, consider integrating a more comprehensive review of existing studies on plant-based beverages. This could involve discussing key findings, trends, and debates in the field to provide context for the study.
Materials and Methods
Provide a more detailed explanation of the sampling methodology, including how participants were recruited, the inclusion criteria, and any sampling biases that may have been present.
Provide a clearer description of the two-part questionnaire used in the study, including the specific questions asked in each section and how they were developed or adapted from existing literature.
Provide a more comprehensive overview of the statistical analysis methods used, including structural equation modeling, Spearman correlation coefficient, and multiple regression analysis. Additionally, explain why these specific methods were chosen and how they were implemented in the study.
Results and Discussion
It would be helpful for the authors to provide context for their findings by discussing relevant literature or theoretical frameworks.
After presenting the data, the authors should discuss the implications of their findings. They should explain how the results contribute to existing knowledge in the field, discuss any practical implications for consumers or policymakers, and identify potential areas for future research.
Throughout the Results section, the authors should highlight key findings or trends to guide readers through the data.
Conclusion
While the authors have effectively summarized the findings, they could emphasize any novel or unexpected results that emerged from the study. Specifically, they should further discuss the inconsistency found regarding the influence of income level on buying behavior for plant-based beverages compared to existing literature.
The authors should provide a clearer discussion of the implications of their findings for plant milk marketers. They should explain how the identified factors influencing attitudes and buying behavior can inform marketing strategies, particularly in addressing health concerns and environmental considerations associated with dairy consumption.
Building on the discussion of implications, the authors could provide practical recommendations for plant milk marketers based on their findings. This could involve suggesting specific approaches for integrating environmental protection awareness into marketing campaigns and positioning plant-based beverages as complementary alternatives to dairy products.
The authors have appropriately acknowledged the limitations of the study, including the reliance on online data collection, the educational background of participants, and the limited scope of variables examined. However, they could expand on these limitations by discussing potential implications for the generalizability of the findings and suggesting avenues for future research to address these limitations.
In addition to discussing limitations, the authors should explicitly call for further research to address the gaps identified in the study. They could emphasize the need for larger-scale studies with more diverse samples and a broader range of variables to provide a comprehensive understanding of consumer behavior regarding plant-based beverages.
Author Response
Dear Reviewer,
Thank you for your valuable contributions and comments. We have tried to improve our study by taking your suggestions into consideration.
Kind regards,

Reviewer 2 Report
Comments and Suggestions for Authors
Manuscript 3023970
Journal Foods
Title Driving Factors Influencing the Decision to Purchase Plant-Based Beverages: a Sample from Türkiye
The manuscript entitled “Driving Factors Influencing the Decision to Purchase Plant-Based Beverages: a Sample from Türkiye” describes the determinants of buying behaviour related to plant-based beverages. The findings revealed that individual’s sociodemographic characteristics and environmental protection are the main factors. The manuscript is interesting but several parts need improvement. See the comments in the file.

Moderate changes are necessary
Author Response

(The authors gave the same response as above.)

Reviewer 3 Report
Comments and Suggestions for Authors
1.How do you distinguish the objective and subjective knowledge?
2.We suggest revising your H2 as …. Toward buying plant-based beverages
3.What is your definition of trust?
4.What is your definition of consumer’s attitude?
5.Would your H12g~H12i be overlapped with H1? Or how do you distinguish them?
6.Please show the questionnaire since we are not clear whether your questions to participants related to your hypotheses or not.
7.We suggest having comparison between the results from the group of purchased plant-based beverages vs the group of not purchased.
8.We wonder taste (that shown 100% in your table 2) will be in which construct that in your table 3?
9.You had mentioned [As the scores of animal welfare and environmental protection among the study factors increased, attitude scores increased by 6.7% and 8.6%, respectively.], please clarify the attitude scores increased from what. We failed to find this evidence anywhere.
10.Again, we failed to find that taste play a role in your Figure 1. Your data had shown that participants choose taste and health benefits are key factors while you did not put them into the model that you tried to set up.
11. We will suggest adding the discussion on the part that your finding is inconsistent with literature. This will be positive for further studies.
Author Response

(The authors gave the same response as above.)

Round 2
Reviewer 2 Report
Comments and Suggestions for Authors
Authors addressed large part of the comments. I suggest to revise the English language with a native speaker.
Comments on the Quality of English LanguageModerate changes are necessary
Author Response
Dear reviewer,
We have shared our response to your comment in the attached file. Thank you for your contributions.
Kind regards,
